# Cox-Hawkes: doubly stochastic spatiotemporal Poisson processes

**Xenia Miscouridou**[*]                                        *x.miscouridou@imperial.ac.uk*
*I-X and Department of Mathematics, Imperial College London*

**Samir Bhatt**                                                 *s.bhatt@imperial.ac.uk*
*Department of Public Health, University of Copenhagen,*
*School of Public Health, Imperial College London*

**George Mohler**                                              *gmohler@iupui.edu*
*Department of Computer Science, Boston College*

**Seth Flaxman**[†]                                            *seth.flaxman@cs.ox.ac.uk*
*Department of Computer Science, University of Oxford*

**Swapnil Mishra**[†]                                          *swapnil.mishra@nus.edu.sg*
*Saw Swee Hock School of Public Health and Institute of Data Science, National University of Singapore and National University Health System*

[*] *Corresponding author*
[†] *Equal Contribution*

**Reviewed on OpenReview:** *https://openreview.net/forum?id=xzCDD9i4IZ*

## Abstract

Hawkes processes are point process models that have been used to capture self-excitatory behaviour in social interactions, neural activity, earthquakes and viral epidemics. They can model the occurrence of the times and locations of events. We develop a new class of spatiotemporal Hawkes processes that can capture both triggering and clustering behaviour and we provide an efficient method for performing inference. We use a log-Gaussian Cox process (LGCP) as prior for the background rate of the Hawkes process which gives arbitrary flexibility to capture a wide range of underlying background effects (for infectious diseases these are called endemic effects). The Hawkes process and LGCP are computationally expensive due to the former having a likelihood with quadratic complexity in the number of observations and the latter involving inversion of the precision matrix which is cubic in observations. We propose a novel approach to perform MCMC sampling for our Hawkes process with LGCP background, using pre-trained Gaussian process generators which provide direct and cheap access to samples during inference. We show the efficacy and flexibility of our approach in experiments on simulated data and use our methods to uncover the trends in a dataset of reported crimes in the US.

*Keywords:* Gaussian process, self-excitation, clustering, Bayesian inference

# 1 Introduction

Hawkes processes are a class of point processes that can model self or mutual excitation between events, in which the occurrence of one event triggers additional events, for example: a violent event in one geographical area on a given day encourages another violent event in an area nearby the next day. A unique feature of Hawkes processes is their ability to model exogenous and endogenous "causes" of events. An exogenous cause happens by the external addition of a event, while endogenous events are self-excited from previous events by a triggering kernel. An example of the difference between these two mechanisms is in disease transmission - an exogenous event could be a zoonosis event such as the transmission of Influenza from birds, while endogenous events are subsequent human to human transmission. Due to their flexibility and mathematical tractability, Hawkes processes have been extensively used in the literature in a series of applications. They have modelled among others, neural activity (Linderman et al., 2014), earthquakes (Ogata, 1988), violence (Loeffler & Flaxman, 2018; Holbrook et al., 2021) and social interactions (Miscouridou et al., 2018).

The majority of research on Hawkes processes focuses on the purely temporal settings where events occur and are subsequently triggered only in time. However, many practical problems require the inclusion of a spatial dimension. This inclusion is motivated by several factors, first, natural phenomena that self-excite tend to do so both spatial and temporally e.g. infectious diseases, crime or diffusion over a network. Second, natural processes tend to cluster closely in space and time (Tobler, 1970). Third, in parametric formulations residual variation persists and this is often structured in both space and time (Diggle & Ribeiro, 2007). A wide body of research exists in modelling spatial phenomena ranging from Kriging (Matheron, 1962) to model based estimates (Diggle & Ribeiro, 2007) using Gaussian processes. In the more general Gaussian process, which provides a prior function class, spatial phenomena are modelled through a mean function and a covariance function that allows control over the degree of clustering as well as the smoothness of the underlying functions. Specifically for applications for spatial point patterns, an elegant formulation using log-Gaussian Cox processes (LGCP), (Møller et al., 1998) is commonly used (Diggle et al., 2013). LGCPs can capture complex spatial structure but at a fundamental level are unequipped with a mechanism to model self-excitation. When examining the processes' endogenous and exogenous drivers, the lack of a self-exciting mechanism can potentially lead to spurious scientific conclusions even if prediction accuracy is high. For example, appealing again to the Influenza example, only modelling the distribution of cases using an LGCP will ignore the complex interplay of zoonosis events and secondary transmission events, both of which require different policy actions.

The inclusion of space has a long history via the Hawkes process triggering mechanism - fistly modelled using the Epidemic Type Aftershock Sequence (ETAS) kernel (Ogata, 1988) but many subsequent approaches now exist. However, to our knowledge, very few approaches consider spatial and temporal events in *both* the exogenous and endogenous Hawkes process mechanisms - that is where events can occur in space and time, and then these events trigger new events also in space and time. Many mechanisms have been proposed for space-time triggering kernels (Reinhart, 2018), but it is not clear nor straightforward how to also allow for exogenous space-time events simultaneously. In the vast majority of previous applications, exogenous events occur at a constant rate in both space and time. Non constant approaches exist but have usually been case specific. For example the use of a periodic function had been effective in modelling seasonal malaria (Unwin et al., 2021). Some studies do provide nonparametric approaches for the background rate: Lewis & Mohler (2011) provide an estimation procedure for the background and kernel of the Hawkes process when no parametric form is assumed for either of the two. Donnet et al. (2020) and Sulem et al. (2021) use nonparametric estimation on the Hawkes kernel whereas Miscouridou et al. (2018) use a nonparametric prior on the background based on completely random measures to construct Hawkes processes that build directed graphs.

Other recent approaches use neural networks to estimate the rate but the majority of these approaches are in purely temporal settings such as Omi et al. (2019). There are a few that consider marked point processes such as Du et al. (2016). However, a marked temporal point process which can use marks as locations would not capture the spatial correlations we are interested in. More recently Zhou et al. (2022) and Chen et al. (2021) provided a way to model spatiotemporal settings with neural networks. However, both of them still lack the ability to capture a more flexible background where background intensity is not just a constant

but by itself produces clustering in time and space (we like to emphasise here that this type of clustering is different from the one emerging from the self-exciting nature of the process).

Beyond neural network approaches, there exist non deep models that deal with non-linear intensities. An example is Zhou et al. (2020) who proposed a sigmoid Gaussian Hawkes process with baseline intensity and triggering kernel drawn from a Gaussian process prior, passed through a sigmoid link function to guarantee non-negativity. Similarly, Apostolopoulou et al. (2019); Sulem et al. (2021); Malem-Shinitski et al. (2021) propose point process models with a non-linear component that allows both excitatory and inhibitory relationships in continuous time.

Here we propose a novel space-time approach that combines Hawkes processes (Hawkes, 1971) with log-Gaussian Cox processes (Møller et al., 1998; Diggle et al., 2013). This synthesis allows us, for the first time, to have an exogenous background intensity process with self-excitation that is stochastic and able to vary in both space and time. We provide a suite of new methods for simulation and computationally tractable inference. Our methods leverage modern computational techniques that are scalable and can efficiently learn complex spatiotemporal data. We apply our approach on both simulated and real data. Our novel addition of an LGCP prior in both space and time is accompanied with new computational challenges: a Hawkes process is quadratic in complexity due to a double summation in the likelihood, and LGCPs incur cubic complexity from matrix inversions. To ensure our approach is scalable and still competitive with standard Hawkes processes we utilise a recently developed Gaussian process approximation (Mishra et al., 2022; Semenova et al., 2022) that obliviates the need for repeated matrix inversions. Our work represents a step towards more general, scalable, point process framework that encodes more flexible and plausible mechanisms to represent natural and physical phenomena.

**Our contributions**

A summary of the contributions of our work is: (i) We provide a novel model formulation for a highly flexible self-exciting process that can capture endogenous and exogenous events in both space and time. Our utilisation of LCGPs for the exogenous background rate is extremely flexible and follows from the current state-of-the-art in spatial statistics (Diggle et al., 2013). (ii) In contrast to previous work such as Loeffler & Flaxman (2018), our framework admits a generative model that can produce stochastic realisations at an arbitrary set of locations. We provide a novel algorithm to sample from this generative process. (iii) We offer an efficient Bayesian inference approach that ensures our more flexible model is still as scalable as standard Hawkes processes and straightforward to implement computationally. (iv) Our framework is directly applicable to numerous spatiotemporal problems where there are both endogenous and exogenous causes e.g. for natural or social phenomena such as crime, diseases, environment, or human behaviour.

## 2 Related methods

As mentioned before, modelling space through Hawkes processes was first used with the Epidemic Type Aftershock Sequence (ETAS) kernel (Ogata, 1988) and other approaches followed some of which exist in Reinhart (2018). For modelling spatial point patterns without self-excitation, log-Gaussian Cox processes (LGCP) Møller et al. (1998) provide an elegant approach as explained in Diggle et al. (2013).

Reinhart (2018) provide an overview on spatiotemporal Hawkes processes explaining various options for the form of the intensity, the kernels and the corresponding simulating algorithm. However, the case of an LGCP background is not discussed in the review.

Our approach is the first to use an LGCP to capture the background underlying effects (these are called endemic effects in infectious disease modelling but here we will use this term broadly for other applications too) and can model the exact spatial and time locations.

Loeffler & Flaxman (2018) aim to understand whether gun violence in Chicago is contagious or merely clusters in space and time. To this end, they use a spatiotemporal Hawkes model and a space-time test to distinguish between the two. The model uses a kernel density estimator for the background (endemic) effects and a kernel for the epidemic events that is separable in space and time. Their model has a different

construction as it does not admit a generative procedure since the background rate is estimated using kernel density estimators.

Similarly to Loeffler & Flaxman (2018), Holbrook et al. (2021) build a scalable inference algorithm for parametric spatiotemporal self-exciting processes. Their proposed model is the one of Loeffler & Flaxman (2018) which is based on a Gaussian kernel smoother for the background. The main contribution is to overcome the bottleneck of the quadratic computational complexity of such a point process. The authors develop a high-performance computing statistical framework to do Bayesian analysis with Metropolis-Hastings using contemporary hardware. They apply it on a gunfire dataset which covers a larger dataset and more fine-grained than the one in Loeffler & Flaxman (2018).

The combination of a Hawkes process with an LGCP is found in Linderman & Adams (2015) where the authors propose a purely temporal multivariate Hawkes process with LGCP in the background with the goal to infer a latent network structure given observed sequences of events. This approach is based on Linderman & Adams (2014) but in discrete time and with an improved inference scheme based in mini batches. However both of these two have different scope to our work and work only with temporal data.

Finally, Mohler (2013) develops a purely temporal Hawkes process model with LGCP background for count (aggregated) events. Mohler (2013) builds a Metropolis adjusted Langevin algorithm for estimation and uses the algorithms to disentangle the sources of clustering in crime and security data. We are instead interested in modelling and predicting exact event times and locations.

Few neural network models (Du et al., 2016; Mei & Eisner, 2017; Omi et al., 2019; Zhang et al., 2022) have been suggested but they are only temporal in nature. A few of them provide a way to model a spatial domain, by treating space locations as discrete marks. However, using marks doesn't allow us to model spatial correlations, as observed in many settings. Okawa et al. (2019)) extend neural networks to spatial settings but lack the ability to predict the next event in space and time. Work most similar to ours is Chen et al. (2021) and Zhou et al. (2022), where they apply neural network inspired point process models for spatiotemporal data. However, they still lack the ability to account for anything more than a constant background intensity.

## 3 Model

### 3.1 Point process intensity

A Hawkes process is an inhomogeneous Poisson point process defined in terms of a counting measure and an intensity function or rate. For a generic spatiotemporal inhomogeneous Poisson point process on the domain $\mathcal{X} \times [0, T)$, for $\mathcal{X} \subset \mathbb{R}^d$ we denote the counting measure of the process by $N$ and the conditional intensity by $\lambda$. The definition of a generic inhomogeneous point process intensity in space and time is as below. For $\mathbf{s} \in \mathcal{X} \subset \mathbb{R}^d$ (generally $d$ here represents Euclidean or Cartesian coordinates) and $t \in [0, T)$

$$\lambda(t, \mathbf{s}) = \lim_{\Delta t, \Delta \mathbf{s} \to 0} \frac{\mathbb{E}[N[(t, t + \Delta t) \times B(\mathbf{s}, \Delta \mathbf{s})] \,|\, \mathcal{H}_t]}{\Delta t \times |B(\mathbf{s}, \Delta \mathbf{s})|} \tag{1}$$

where $\mathcal{H}_t$ denotes the history of all events of the process up to time $t$, $N(A)$ is the counting measure of events over the set $A \subset \mathcal{X} \times [0, T)$ and $|B(\mathbf{s}, \Delta \mathbf{s})|$ is the Lebesgue measure of the ball $B(\mathbf{s}, \Delta \mathbf{s})$ with radius $\Delta \mathbf{s} > 0$. The intensity $\lambda$ has to be non-negative, i.e. $\lambda \geq 0$. Note that the spatial locations can be univariate, referring for example to regions or countries, or bivariate such as geographical coordinates of longitude and latitude or even multivariate depending on the context.

### 3.2 Hawkes process intensity

Hawkes processes were originally proposed by Hawkes (1971) as temporal point processes. The intensity is conditional on the history of the process such that the current rate of events depends on previous events. We focus on self-exciting Hawkes processes, in which historic events encourage the appearance of future events. We develop spatiotemporal self-exciting processes which can predict the rate of events happening at specific

locations and times. The conditional intensity defined as in equation 1 admits the form

$$\lambda(t, \mathbf{s}|\mathcal{H}_t) = \mu(t, \mathbf{s}) + \sum_{i:t_i<t} g\left(t - t_i, \mathbf{s} - \mathbf{s}_i\right),\tag{2}$$

where $(t_1, t_2, \ldots, t_n)$ denotes the ordered sequence of the times of the observed events and $(\mathbf{s}_1, \mathbf{s}_2, \ldots, \mathbf{s}_n)$ their corresponding spatial locations. Events arise either from the background rate $\mu(t, \mathbf{s})$ (exogenous or non excitation effects) or from the triggering function/kernel $g$ (endogenous or self-excitation effects). $\mu$ is non-negative to ensure that the initial intensity is non-negative and we take $g$ non-negative as we consider excitation effects and do not deal with inhibition. For the scope of our work, we are interested in excitation, however for other applications such as neural connectivity patterns where inhibition is needed, one can read for example Cai et al. (2022).

$g$ can be parametric or it can be estimated using full nonparametric assumptions, as done for example in Donnet et al. (2020); Sulem et al. (2021). Similarly, it can take a form of separable (additive or multiplicative) or non-separable kernels in space and time. There exists a lot of work covering all these cases for purely temporal processes but not in spatiotemporal settings. To give some background, in purely temporal cases there exist guarantees on the estimation and under certain conditions there are consistency results as well identifiability results. However once we add the spatial component, the results do not necessarily extend. Therefore, we consider here the simple case of a separable form of a product of an exponential kernel in time and a Gaussian kernel in space. For multivariate linear purely temporal Hawkes processes with constant background, it is a known result that one can recover the parameters, i.e. the process is identifiable and one can also prove consistency results: Donnet et al. (2020) prove this and give posterior concentration rates. Some results also exist for non-linear Hawkes processes with constant backgrounds: Brémaud & Massoulié (1996) provide results on the uniqueness of the stationary solution but they do not study estimation of the parameters. Similarly Sulem et al. (2021) study general non-linear and nonparametric Hawkes processes and provide conditions on the Bayesian methods to estimate the parameters with a (presumably optimal) concentration rate. Other approaches with time-varying backgrounds exist (e.g. Unwin et al. (2021); Zhou et al. (2020)) but there are no theoretical results that apply directly in that case (linear or non-linear). We think it is an interesting direction to study the theoretical properties of spatiotemporal hawkes processes and we would encourage research in that direction. We would like to note though that in any Hawkes process with a temporally or spatiotemporally varying background, stationarity is not relevant anymore as the background is changing in time and therefore the expectation of the intensity cannot be constant.

The process with intensity defined in equation 2 can be treated as a Poisson cluster process, with mean number of offsprings given by $b = \int_\mathcal{X} \int_0^\infty g(dt, d\mathbf{s})$. To ensure that the cluster sizes are almost surely finite, we require that $b \in (0, 1)$ as each generation of offsprings follows a geometric progression, with expected total cluster size of $\frac{1}{1-b}$. For $b = 0$ we have a Cox process where as for $b \geq 1$ the process explodes and we would have an infinite number of events in finite time. To see how explosion emerges, we refer the reader to of (Grimmett & Stirzaker, 2001, Chapter 5) which give the calculations on the expected number of descendants of one atom. More on the implications of the values of $b$ can be found in Asmussen & Glynn (2003).

The triggering function $g$, centered at the triggering event, is the intensity function for the offspring process. Properly normalised, it induces a probability distribution for the location and times of the offspring events. The cluster process representation of the Hawkes process (Hawkes & Oakes, 1974) will prove crucial to the efficient simulation of self-exciting processes which we give in section 4.1.

We give below the triggering kernel that admits a separable form of a product of an exponential kernel in time and a Gaussian kernel in space. Both of these choices are relevant for the applications we consider in the paper as we know from previous work Loeffler & Flaxman (2018); Holbrook et al. (2021) that the decay and variation in crime data can can be well explained by the decay prescribed by an exponential and Gaussian kernel respectively. For $t > 0$ and $\mathbf{s} \in \mathcal{X} \subset \mathbb{R}^d$ the self-exciting part of the rate is given by

$$g(t, \mathbf{s}) = \alpha\beta \exp\left(-\beta t\right) \frac{1}{\sqrt{2\pi|\Sigma|}} \exp\left(-\mathbf{s}^T\Sigma^{-1}\mathbf{s}\right),\tag{3}$$

where $\alpha > 0, \beta > 0$ and $\Sigma$ a semi-positive definite matrix.

For the temporal part we use the widely used exponential kernel, originally proposed by Hawkes (1971), giving exponential decay which is suitable for the applications we are interested in. Note that an exponential kernel is not always a reasonable choice, for instance in infectious diseases one would prefer the Rayleigh kernel (e.g. see Unwin et al. (2021)). For the spatial part, we use a Gaussian kernel which is suitable for modelling spatial locations especially for social violence settings as first proposed and used by other authors in literature. Specifically, (Loeffler & Flaxman, 2018; Holbrook et al., 2021) analyse the public policy and crime implications of a larger gunshot dataset which includes the data used in this paper and choose these forms for the kernels.

As mentioned above we consider here a separable form of $g$. Note that non-separable kernel approaches exist in literature, such as Jun & Cook (2022), in which temporal patterns differ according to location. However one cannot naively use those without properly assessing identifiability concerns. The current construction can be extended to cover those, however this would be beyond the scope of the novelty of the current method and the scope of the applications we consider here.

The other part of the intensity is $\mu(t, \mathbf{s})$, which is the background rate of the process. It is a nonnegative function with initial nonzero value that captures the underlying patterns in space and time that encourage the clustering of events in those time and space locations. It often takes the form of a constant for simplicity, or a parametric form such as periodic as assumed in Unwin et al. (2021) or can even have a nonparametric prior constructed on random measures as in Miscouridou et al. (2018). As further explained in more detail below, we assume a log-Gaussian process prior on $\mu(t, \mathbf{s})$ which to our knowledge has not been used before in the literature of spatiotemporal Hawkes processes.

### 3.3 Latent log Gaussian process for background rate

We use a latent Gaussian process ($\mathcal{GP}$) to determine the background rate of events in time $t \in R$ and space $\mathbf{s} \in R^d$. This means that the background rate takes the form

$$\mu(t, \mathbf{s}) = \exp\left(f\left(t, \mathbf{s}\right)\right) \tag{4}$$

where $f(t, \mathbf{s})$ is a function realisation from a Gaussian process prior in space and time. Formally, a Gaussian process is a collection of random variables, such that any finite collection of them is Gaussian distributed. $\mathcal{GP}$s are a class of Bayesian nonparametric models that define a prior over functions which in our case are functions over time and space. Similarly to a probability distribution that describes random variables which are scalars or vectors (for multivariate distributions), a Gaussian process is distribution over functions and belongs in the family of stochastic processes.

$\mathcal{GP}$s are a powerful tool in machine learning, for learning complex functions with applications in regression and classification problems. We refer the reader to (Rasmussen & Williams, 2005, Chapter 2) for details on Gaussian processes and their properties.

A Gaussian process on $\mathbb{R}^D$, for any $D > 0$ is completely specified by its mean function $m(\cdot)$ and covariance function $k(\cdot, \cdot)$. We will denote a draw from a Gaussian process as

$$f(\cdot) \sim \mathcal{GP}\left(m(\cdot), k(\cdot, \cdot)\right).$$

The Gaussian process is centered around its mean function, with the correlation structure (how similar two points are) of the residuals specified via the covariance kernel. Properties of the underlying function space such as smoothness, differentiability and periodicity can be controlled by the choice of kernel. One of the most popular choices of covariance kernel, and the one we choose to introduce the model with, is the Gaussian kernel (also commonly called the squared exponential kernel), defined for $\mathbf{u}, \mathbf{u}' \in \mathbb{R}^D$ by the covariance function

$$Cov\left(f\left(\mathbf{u}\right), f\left(\mathbf{u}'\right)\right) = k\left(\mathbf{u}, \mathbf{u}'\right) = \omega^2 \exp\left(-\frac{1}{2l^2}|\mathbf{u} - \mathbf{u}'|^2\right) \tag{5}$$

where $|\mathbf{u}|$ denotes the Euclidean norm, i.e. it is equal to $|\mathbf{u}| = \sqrt{\sum_i \mathbf{u}_i^2}$ if $u$ is a vector ($D > 1$ e.g. the spatial locations) and to the absolute value of $\mathbf{u}$ if $\mathbf{u}$ is a scalar ($D = 1$ e.g. timestamps). $\omega^2 > 0$ defines the

kernel's variance scale and $l > 0$ is a length scale parameter that specifies how nearsighted the correlation between pairs of events is. The hyperparameters can be varied, thus also known as free parameters. The kernel and mean of the $\mathcal{GP}$ together fully specify the prior distribution over functions.

We will consider an additive separable kernel with a bivariate spatial dimension $\mathbf{s} = (x, y)$ and univariate temporal dimension $t$. Note that one could naively consider a joint $f_{\mathbf{s}t}$ with no assumptions of additivity (or other form of structure) at all. However this would not be advisable as it would be impossible in this case to guarantee that we can recover the underlying latent functions. When there is not enough structure in the form of the background it is much more difficult to study the identifiability of the latent functions $f_t$ and $f_{\mathbf{s}}$. In non-identifiable cases, the prior dominates the estimation and the estimated $f_t, f_{\mathbf{s}}$ will be heavily influenced by the prior and not by the data. We consider here the additive structure as a minimum type of structure to assume on the background latent process which is still very generic, able to capture arbitrary background trends.

In order to ensure a nonnegative background we exponentiate the additive kernel. From this kernel specification the background intensity $\mu(t, \mathbf{s})$ follows a log-Gaussian Cox process (Møller et al., 1998; Diggle et al., 2013) over space and time

$$\mu(t, \mathbf{s}) = \exp\left(f_{\mathbf{s}}\left(\mathbf{s}\right) + f_t\left(t\right)\right) \tag{6}$$
$$f_t \sim \mathcal{GP}\left(m_t, k_t\right)$$
$$f_{\mathbf{s}} \sim \mathcal{GP}\left(m_{\mathbf{s}}, k_{\mathbf{s}}\right),$$

where $m_t$ and $m_{\mathbf{s}}$ are the $\mathcal{GP}$ mean functions and $k_t$, $k_{\mathbf{s}}$ are the kernels defined by the hyperparameters $\omega_t^2, \omega_{\mathbf{s}}^2, l_t, l_{\mathbf{s}}$.

### 3.4 Full Model Likelihood

To model the spatial coordinates $\mathbf{s} = (x, y)$ and time stamps $t$, we use a Hawkes kernel $g_{t\mathbf{s}}(t, \mathbf{s}) = g_t(t)g_{\mathbf{s}}(\mathbf{s})$ from equation 3 and the log-Gaussian Cox process $\mu(t, \mathbf{s}) = \exp\left(f_{\mathbf{s}}(\mathbf{s} + f_t(t)\right)$ from equation 6. Without loss of generality we will assume here that the Gaussian processes have zero mean. The joint model we consider is a Hawkes process with composite rate $\lambda(t, x, y)$ which is the sum of the intensities of an LGCP process and a Hawkes process

$$\begin{aligned}
\lambda(t, x, y) &= \exp\left(f_{\mathbf{s}}\left(x, y\right) + f_t\left(t\right)\right) \\
&+ \sum_{i:t_i<t} g_t(t - t_i)g_{\mathbf{s}}(x - x_i, y - y_i) \\
&= \exp\left(f_{\mathbf{s}}\left(x, y\right) + f_t(t)\right) \\
&+ \sum_{i:t_i<t} \alpha\beta \exp\left(-\beta(t - t_i)\right) \frac{1}{2\pi\sigma_x\sigma_y} \exp\left(-\frac{(x - x_i)^2}{2\sigma_x^2} - \frac{(y - y_i)^2}{2\sigma_y^2}\right).
\end{aligned} \tag{7}$$

One could see this as an intercept coming from constant contributions by both the temporal and spatial background processes as these are only identifiable through their sum and not separately. Given a set of observed ordered times $(t_1, t_2, \ldots, t_n) \in [0, T]$ and the corresponding locations $(\mathbf{s}_1, \mathbf{s}_2, \ldots, \mathbf{s}_n) \in \mathcal{X}$, let $D$ denote the full data $D = \{t_i, \mathbf{s}_i\}_{i=1}^n$ and $L(D)$ the likelihood. Following equation (7.1.2) in (Daley & Vere-Jones, 2008, Chapter 7) the likelihood is given by

$$\begin{aligned}
L(D) &= \left[\prod_{i=1}^n \lambda(t_i, \mathbf{s}_i)\right] \exp\left(-\int_{\mathcal{X}} \int_0^T \lambda(t, \mathbf{s})dtd\mathbf{s}\right) \\
&= \left[\prod_{i=1}^n \lambda(t_i, x_i, y_i)\right] \exp\left(-\int_{\mathcal{X}} \int_0^T \lambda(t, x, y)dtdxdy\right).
\end{aligned} \tag{8}$$

We give below details on how to simulate from the process with the rate defined in equation 7 and how to perform Bayesian inference using the likelihood from equation 8.

## 4 Methods

### 4.1 Simulation

By construction our model admits a generative process facilitating simulation. This is an important and nuanced advantage over previous spatiotemporal models (Loeffler & Flaxman, 2018; Holbrook et al., 2021) which were not fully generative due to a deterministic parameterisation of the exogenous component. Note that the model of Mohler (2013) does admit a generative model but only for a purely temporal model for aggregated (count) data. In general, Hawkes processes can be simulated in two ways: through an intensity based approach or a cluster based approach. We give below Algorithm 1 to simulate from our model via the latter approach, i.e. through simulating the background first and then the generations of subsequent offsprings. Note that for the hyperparameters $l_t, l_\mathbf{s}, \omega_t^2, \omega_\mathbf{s}^2$ one can either fix them to a known value or (hyper)priors on them.

---

**Algorithm 1** Cluster based generative algorithm for Hawkes process simulation

---

**Require:** Fix $T > 0, \mathcal{X}$
    **Draw** $l_t, l_\mathbf{s} \sim p^+(\cdot)$
    **Draw** $\omega_t^2, \omega_\mathbf{s}^2 \sim p^+(\cdot)$
    **Draw** $a_0 \sim p(\cdot)$
    **Draw** $f_t \sim \mathcal{GP}(0, k_t), f_\mathbf{s} \sim \mathcal{GP}(0, k_\mathbf{s})$
    **Set** $\mu(t, \mathbf{s}) = \exp\left(f_t(t) + f_\mathbf{s}(\mathbf{s}) + a_0\right)$
    **Draw** $N_0 \sim \text{Pois}\left(\int_0^T \int_\mathcal{X} \mu(t, \mathbf{s}) \, dt d\mathbf{s}\right)$
    **Draw** $\left(t_i^{(0)}, \mathbf{s}_i^{(0)}\right)_{i=1}^{N_0}$ from a Poisson Process with rate $\mu(t, \mathbf{s})$ where $s_i^{(0)} = \begin{pmatrix} x_i^{(0)} \\ y_i^{(0)} \end{pmatrix}$
    **Set** $G_0 = \left(t_i^{(0)}, \mathbf{s}_i^{(0)}\right)_{i=1}^{N_0}, \ell = 0$
    **while** $G_\ell \neq \emptyset$ **do**
        **for** $i = 1$ to $N_\ell$ **do**
            Draw $C_i \sim \text{Pois}\left(\int_0^T \int_\mathcal{X} g_{t\mathbf{s}}(t, \mathbf{s}) dt d\mathbf{s}\right)$, the number of offsprings of event $i$
            **for** $j = 1$ to $C_i$ **do**
                Draw $t_j^{(\ell+1)} \overset{iid}{\sim} \text{Exp}(\beta) + t_i^{(\ell)}$,
                Draw $s_j^{(\ell+1)} \overset{iid}{\sim} \text{Normal}\left(\begin{pmatrix} x_i^{(l)} \\ y_i^{(l)} \end{pmatrix}, \begin{pmatrix} \sigma_x^2 & 0 \\ 0 & \sigma_y^2 \end{pmatrix}\right)$
                Set $O_j = \left(t_i^{(\ell+1)}, \mathbf{s}_i^{(\ell+1)}\right)$
            **end for**
        **end for**
        $\ell + = 1$
        $G_\ell = \{\bigcup_{i=1}^{N_\ell} O_1, \ldots, O_{C_i}\}_{\{i: t_i^{(\ell)} < T\}}$
    **end while**
**return** $\bigcup_\ell G_\ell$

---

We use a clustering approach (Hawkes & Oakes, 1974) for simulation which makes use of the Poisson cluster process where each event has a mean number of offspring $b$ (see section 3.2) and relies on the following idea: for each immigrant $i$, the times and locations of the first-generation offspring arrivals given the knowledge of the total number of them are each i.i.d. distributed. We provide the simulation in Algorithm 1. In Algorithm 1 we introduce $a_0$ to denote the total mean of the background rate as the $\mathcal{GP}$s have zero mean. $p^+(\cdot)$ refers to a probability distribution on the positive real line and $p(\cdot)$ a distribution on the real line. As a test check for making sure that our Hawkes process simulations are correct we employ an approximate Kolmogorov-Smirnov type of test adapting Algorithm 7.4.V from Daley & Vere-Jones (2008).

To simulate from our model proposed above, i.e. a Cox-Hawkes process we need to draw from a $\mathcal{GP}$. Since $\mathcal{GP}$s are infinitely dimensional objects, in order to simulate them we have to resort to finite approximations. The most common approach is to implement them through finitely-dimensional multivariate Gaussian distributions. This is the approach we take as well for simulating our $\mathcal{GP}$s. In order to sample points from the LGCP background of the process, we draw an (approximate) realisation from the $\mathcal{GP}$ prior and then use rejection sampling to sample the exact temporal and spatial locations. The algorithm can be found in Appendix A.1.

### 4.2 Inference

Given a set of $n$ observed data $\mathcal{D} = \{t_i, \mathbf{s}_i\}_{i=1}^n$ over a period $[0, T]$ and a spatial area denoted by $\mathcal{X}$, we are interested in a Bayesian approach to infer the parameters and hyperparameters of the model. Denote by $\theta$ and $\phi$ set of the parameters of the background rate $\mu(t, \mathbf{s})$ and the triggering rate $g(t, \mathbf{s})$ respectively. This gives $\theta = (a_0, f_t(t), f_{\mathbf{s}}(\mathbf{s}))$ and $\phi = (\alpha, \beta, \sigma_x^2, \sigma_y^2)$.

The posterior is then given by

$$\begin{aligned} \pi(\phi, \theta | D) &\propto \pi(\theta) \times \pi(\phi) \times L(\mathcal{D}) \\ &= \pi(f_t(t))\pi(f_{\mathbf{s}}(\mathbf{s}))\pi(a_0) \times \pi(\alpha)\pi(\beta)\pi(\sigma_x^2)\pi(\sigma_y^2) \times L(\mathcal{D}). \end{aligned} \tag{9}$$

where $L(\mathcal{D})$ is the likelihood defined in equation 8 and with some abuse of notation, we use $\pi$ to denote both prior and posterior for all the parameters. For $a_0$ we use a Normal prior. The prior for $\pi(\alpha)$, $\pi(\beta)$, $\pi(\sigma_x^2)$ and $\pi(\sigma_y^2)$ is Truncated Normal (restricted on the positive real line) to ensure the positivity of these parameters. For the experiments we demonstrate below we have also tried Gamma and Exponential priors for $\alpha$ and $\beta$, however in that case the MCMC chains showed worse mixing in comparison to the case of the Truncated Normal prior. Note that the prior on the functions $f_t$ and $f_{\mathbf{s}}$ can be further defined by the priors on the hyperparameters $l_t \sim \text{InverseGamma}$, $\omega_t^2 \sim \text{LogNormal}$ for the temporal process and $l_{\mathbf{s}} \sim \text{InverseGamma}$, $\omega_{\mathbf{s}}^2 \sim \text{LogNormal}$ for the spatial.

Our objective is to approximate the total posterior distribution $\pi(\phi, \theta | D)$ using MCMC sampling. A classical Hawkes process has quadratic complexity for computing the likelihood. Only in special cases such as that of a purely temporal exponential kernel the complexity is reduced from quadratic to linear as it admits a recursive construction. See Dassios & Zhao (2013) for an explanation. Note however we cannot apply this in our case as it does not hold when we add (on top of the temporal exponential kernel) the Gaussian spatial kernel. Inference is in general cumbersome and people tend to either resort to approximations or high performance computing techniques such as Holbrook et al. (2021).

A naive formulation of combining log-Gaussian cox processes as the background intensity function in the spatiotemporal Hawkes process will increase the computational complexity for the inference. This happens because in addition to the quadratic complexity arising from the triggering kernel the exogeneous formulation naively introduces a cubic complexity for a LGCP (Diggle et al., 2013).

We propose to circumvent the computational issues through a reduced rank approximation of a Gaussian process (Semenova et al., 2022) through variational autoencoders (VAE). This approach relies on pre-training a VAE on samples from a Gaussian process to create a reduced rank generative model. Once this VAE is trained, the decoder can be used to generate new samples for Bayesian inference. More specifically, in this framework one should first train a VAE to approximate a class of $\mathcal{GP}$ priors (the class of $\mathcal{GP}$ priors learned varies from context to context depending on our prior belief about the problem space) and then utilises the trained decoder to produce approximate samples from the $\mathcal{GP}$. This step reduces the inference time and complexity as drawing from a standard normal distribution $z \sim \mathcal{N}(0, \mathbb{I})$ with uncorrelated $z_i$ is much more efficient than drawing from a highly correlated multivariate normal $\mathcal{N} \sim (0, \Sigma)$ with dense $\Sigma$. For more details see section 2.5 in Semenova et al. (2022). Here we will denote this approximation to the Gaussian

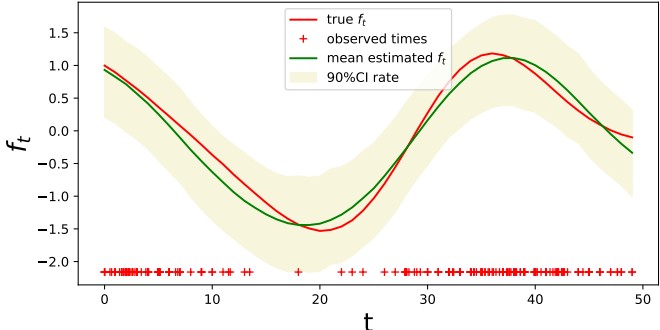

Figure 1: Plot for the temporal Gaussian process $f_t(t)$ on simulated data. The red line is the simulated draw of the Gaussian process, the green line is the mean posterior and the yellow shaded area is the 90% credible interval. The red marks on the x-axis are the exact simulated times from the background process.

Process prior by $\tilde{\pi}$. Hence, we obtain overall the Bayesian hierarchical model

$$
\begin{aligned}
\pi(\phi, \theta | D) &\propto \pi(\theta)\,\pi(\phi) L(D) \\
&= \pi(f_t(t))\pi(f_{\mathbf{s}}(s))\pi(a_0)\pi(\alpha)\pi(\beta)\pi(\sigma_x^2)\pi(\sigma_y^2) \\
&\approx \tilde{\pi}\left(f_t(t)\right)\tilde{\pi}(f_{\mathbf{s}}(s))\pi(a_0)\pi(\alpha)\pi(\beta)\pi(\sigma_x^2)\pi(\sigma_y^2).
\end{aligned}
\tag{10}
$$

The code for simulation and inference for this class of models of Cox-Hawkes processes implemented in python and numpyro (Phan et al., 2019) can be found at `https://github.com/misxenia/Spatiotemporal_Cox_Hawkes`.

## 5 Experiments

We demonstrate the applicability of our methods on both simulated and real data. For simulations our goal is twofold: (i) to show that we can accurately estimate the parameters of both the background and self-exciting components thereby showing that we recover the true underlying mechanisms and (ii) to show that our method performs well under model misspecification, thereby showing our model is sufficiently general to be used in real data situations where the true underlying data generating mechanism is unknown. On real settings we apply our methods to gunfire data used in Loeffler & Flaxman (2018) detected by an acoustic gunshot locator system to uncover the underlying patterns of crime contagion in space and time. We show how our model can be used as an actionable tool by practitioners to understand and measure contagion effects in important settings. Note that throughout the following we refer to the model used for simulating data as the true model.

### 5.1 Experiment 1: Simulated Data

We simulate data from a Hawkes process with rate as given in equation 7 on the domain $[0, T] = [0, 50], \mathcal{X} = [0, 1] \times [0, 1]$. For the background rate which governs the exogenous events we simulate a realisation of the latent (separable) spatiotemporal Gaussian process with covariance kernels defined as in equation 5 using $l_t = 10, \omega_t^2 = 1, l_{\mathbf{s}} = 0.25, \omega_{\mathbf{s}}^2 = 1$. The simulated $f_t(t)$ from the temporal Gaussian process can be seen in Figure 1 in red and the temporal events drawn from this background are also shown in red on the x-axis. The simulated two-dimensional spatial Gaussian process can be seen at the left plot of Figure 2. Note that we also use an explicit constant intercept of $a_0 = 0.8$ giving an overall background rate of $\exp(a_0 + f_t + f_{\mathbf{s}})$ and in inference we use a Normal prior on it.

For the diffusion effect we use a triggering spatiotemporal kernel of the form in equation 3 with values $\alpha = 0.5, \beta = 0.7$ for the exponential kernel. For the Gaussian spatial kernel we will assume a common

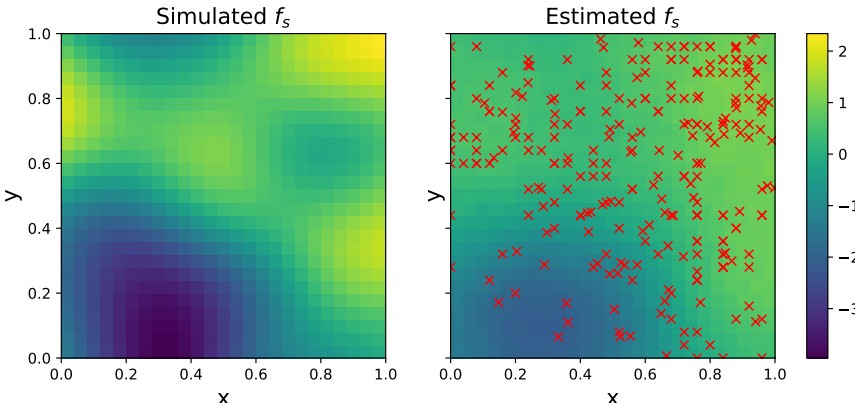

Figure 2: Simulated draw and the posterior predictive distribution for the 2-dimensional spatial Gaussian process. The simulated $f_{\mathbf{s}}(x, y)$ is shown on the left on a regular grid and the mean predictive distribution is shown on the right with the simulated locations in red.

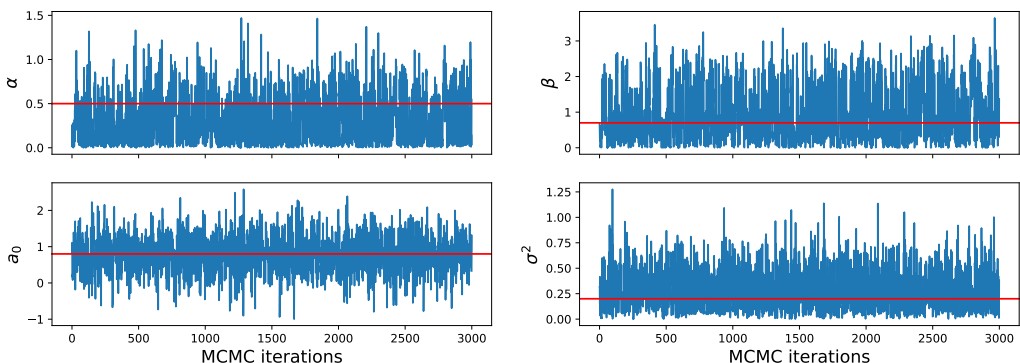

Figure 3: MCMC trace for $a_0, \alpha, \beta, \sigma$ on simulated data where the red line shows the simulated value for the experiments. The samples shown are collected from 3 chains.

parameter $\sigma^2$ for both $\sigma_x^2$ and $\sigma_y^2$ which we will assume to be 0.5. This gives a set of around $n = 210$ spatiotemporal points $\{t_i, x_i, y_i\}_{i=1}^n$ of which the ratio of background to offspring events is roughly $1:1$.

For inference we run 3 chains with $1,500$ samples each of which 500 were discarded as burn in, using a thinning size of 1. In Figure 3 we report the trace plots for the parameters $\alpha, \beta, \sigma$ which define the triggering kernel that governs excitation. We also report $a_0$ which we used as the total mean of the latent Gaussian process $\mu(t, x, y) = \exp\left(f_t(t) + a_0 + f_{\mathbf{s}}(x, y)\right)$. We use a Normal prior on $a_0$. In all cases the simulated values shown in red are within the trace coverage. In our experiments we combine the samples (after removing the warmup iterations) from all the chains. The plots overall show good convergence with good mixing between the chains and no multimodal behaviour. Regarding the Gaussian process fitting, we show the posterior predictive plots in Figures 1 and 2. For the one-dimensional temporal Gaussian process we plot the simulated draw $f_t(t)$ in Figure 1 in red. The blue line is the mean posterior of $f_t(t)$ and the yellow shaded area is the 90% credible interval obtained from the posterior predictive distribution. The red dots on the x-axis are the exact simulated time events drawn from the process. The 90% credible interval covers well the simulated function, and the mean posterior predictive is very close to the simulated one, showing good model fit.

For the two-dimensional spatial Gaussian process we plot the simulated draw function $f_{\mathbf{s}}(x, y)$ at the left plot of Figure 2. The the mean posterior predictive distribution is shown in the centre and the mean predictive

distribution with the true simulated locations embedded on it is shown on the right. The color scale on the right shows the relative values ranging from dark blue (smallest) to yellow (highest). The simulated and the mean of the posterior predictive distribution, are relatively similar when compared visually which shows a good fit for the model.

To quantify convergence we require good $\hat{R}$ diagnostics. For all our results the $\hat{R}$ diagnostics returned by the sampler were in $[1, 1.002]$ for all the estimated parameters. This, combined with visual inspection of the MCMC trace show good evidence of convergence and good mixing behaviour.

To further validate our method we perform a goodness of fit test. From the estimated model parameters we obtain above, we compute the estimated intensity and then apply the time-rescaling theorem (Meyer, 1969; Papangelou, 1972) which states that if the process is correct, the rescaled times are independent exponentially distributed random variables with unit rate. We can thus compare the empirical and theoretical distributions to assess how well the chosen model agrees with the empirical data. We make this comparison using a quantile-quantile (QQ) plot and a Kolmogorov-Smirnov test. We provide in Appendix A.2 in Figure 9 the QQ plot which shows a good fit as the best fit line has a slope of 1.1 and intercept of $-0.2$ with the coefficient of determination that measures the correlation between the two axes to be 0.98. Secondly, we perform hypothesis testing using the Kolmogorov-Smirnov statistic to test the null hypothesis that two distributions were drawn from the same distribution. Our p-value is $\gg 0.05$ suggesting that at a confidence level of 95% we cannot reject the null hypothesis. Given the above, we conclude a good fit of our process to the data.

## 5.2 Experiment 2: Model misspecification

Our second experiment on simulated data compares and contrasts our method (LGCP-Hawkes) to a Hawkes process with constant background and a pure log Gaussian Cox Process. The intensity for our LGCP-Hawkes model is equation 7, for Hawkes it is equation 2 with constant $\mu(t, \mathbf{s}) = \mu$ and for LGCP it is equation 6.

We simulate data from these three inhomogeneous point process models and then fit each model on every dataset on a train set and perform prediction on a test set. Note that we also fit under a homogeneous Poisson model as it's the baseline giving the simplest spatiotemporal model that exists. We show that our model Hawkes-LGCP is a reasonable approach even when there is model mismatch (i.e. when the data are drawn from a pure Hawkes or pure LGCP). It is therefore a good approach to use in real data scenarios when the underling data generating mechanism is unknown.

It is in general challenging to evaluate the quality of model fit from different point process models.

We use two ways to evaluate the model fit and generalisation ability of our model. We first adopt a procedure to predict the temporal and spatial locations of future events under the inference model and then compute the combined error between those events and events generated under the true generating model that was used for simulation. This procedure mirrors the properties that practitioners would desire from their model in real world settings.

The metrics we use to test the generalisation ability of the model are (a) the combined root mean square error (RMSE) between the exact simulated and predicted events and (b) the normalised negative log-likelihood of the test events. The formula we use is the following $RMSE = RMSE_{\mathbf{s}} + RMSE_t$ where $RMSE_t = \sqrt{\frac{1}{n_{test}} \sum_{j=1}^{n_{test}} (t_j - \tilde{t}_j)^2}$ and $RMSE_{\mathbf{s}} = \sqrt{\frac{1}{n_{test}} \sum_{j=1}^{n_{test}} (x_j - \tilde{x}_j)^2 + \frac{1}{n_{test}} \sum_{j=1}^{n_{test}} (y_j - \tilde{y}_j)^2}$ for $t_t, x_i, y_i$ here denoting the true events and $\tilde{t}_i, \tilde{x}_i, \tilde{y}_i$ denoting the predicted ones. The RMSE with its standard error is demonstrated in Figure 4 (top). Similarly we evaluate the normalised negative log-likelihood (NNL) on the test data in each case given the setup explained below and report the mean and standard error in Figure 4 (bottom). The normalisation is given through the division of the original negative log-likelihood by the number of datapoints involved (here the number of test points).

The experimental setup is as follow. We simulate 100 datasets (each of which give on average 300 events) over a fixed time window and a fixed spatial domain and then do a train-test split. We repeat this using as generating model each of the models.

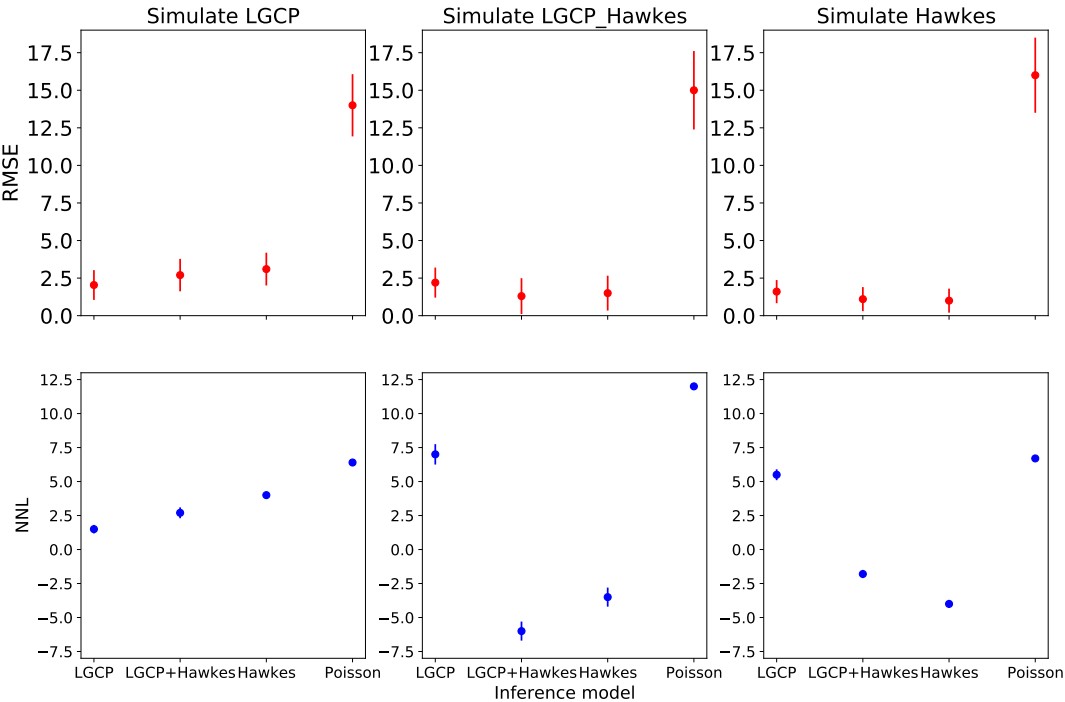

Figure 4: Plots to support the model misspecification experiment. Top: Average RMSE and its standard error (combined for space and time) reported for the model misspecification experiment. Bottom: Average negative normalised log-likelihood and its standard error on test data. The left plot corresponds to a simulated dataset from an LGCP model, the middle to an LGCP-Hawkes and the right from a Hawkes model. In all three cases we perform inference under all LGCP, LGCP-Hawkes, Hawkes as well as Poisson (baseline).

For every dataset, we then perform inference under our MCMC scheme under every model. Given the estimated parameters, we predict 200 times the next 10 future events which we compare to those of the test set. We compute the error between the true and estimated events across the 200 predictions and the 100 simulations. We report the mean and standard error of the RMSE and normalised negative log-likelihood (NNL) graphically in Figure 4. This plot shows how good each model is in predicting the near future.

As shown in Figure 4, and as expected, the RMSE error is always lowest when the true model is used for inference, however in all cases the next best model is LGCP-Hawkes although the differences are not always statistically significant. This provides evidence for our model's ability to flexibly capture a wide range of underlying patterns. Looking at the RMSE error in all cases the worst model is the Poisson baseline, as its constant intensity in space and time cannot capture the inhomogeneities in the data. These results highlight that when the true data generating process is unknown, which is the default scenario in real world settings, our model is likely to be a robust choice. The NNL results suggest the same conclusion, as in all cases NNL is always lowest when the true model is used for inference, and in all cases the next best model is LGCP-Hawkes. The difference with the RMSE results is that under an LGCP-Hawkes model and a Hawkes model the log-likelihood of an LGCP deviates quite a lot from the LGCP-Hawkes and Hawkes model.

Furthermore, what is of interest is to see how the LGCP-Hawkes model under study correctly recovers the other processes, namely Poisson, LGCP and Hawkes. Recall that intensity consists of two parts, the background and the excitation. We will report in each case two ratios, $r_B$: the ratio of the background to

the total intensity and $r_E$: the ratio of the excitation to the total intensity to explain how our model recovers correctly the mechanism which drives the appearance of the events. We will also consider the coefficient for the excitation part of the intensity, which we denote by $c = \alpha \times \beta$ and the parameter of the background, $a_0$ and will report their respective mean estimates.

Applying our LGCP-Hawkes model (that assumes an intensity as in equation 7) on simulated data obtained from the LGCP with true value $a_0 = 0.5$, and using the posterior mean of the parameters, we estimate $r_B = 0.91, r_E = 0.09$. The mean estimates for the excitation coefficient $c$ and $a_0$ are 0.02 and 0.58 respectively. These suggest that the model has correctly inferred that there is negligible contribution to the intensity coming from the self-excitation part, meaning that events arise because of spatiotemporal background variation and not contagion.

Applying our LGCP-Hawkes model on simulated data obtained from the Poisson process with $a_0 = 0.8$, we estimate $r_B = 0.85, r_E = 0.15$. This suggests that most of the contribution comes from the background and a small part from the excitation. The mean estimate for $c$ and $\alpha_0$ are 0.09 and 0.52 respectively. These suggest that the model has correctly inferred that there is very little contribution to the intensity coming from the self-excitation part, meaning that events arise because of spatiotemporal background variation and not contagion.

Applying our LGCP-Hawkes model on simulated data from the Hawkes process with $a_0 = 0.5$ and $c = \alpha \times \beta = 0.4$, we estimate $r_B = 0.43, r_E = 0.57$ which suggests that the events of this process are due to both the background and the self-exciting kernel. The mean estimate for the excitation coefficient $c$ and $\alpha_0$ are 0.42 and 0.52 respectively, which shows that the model has properly recovered the contribution from both background effects and self-exciting behaviour.

### 5.3 Experiment 3: Gunshot Dataset

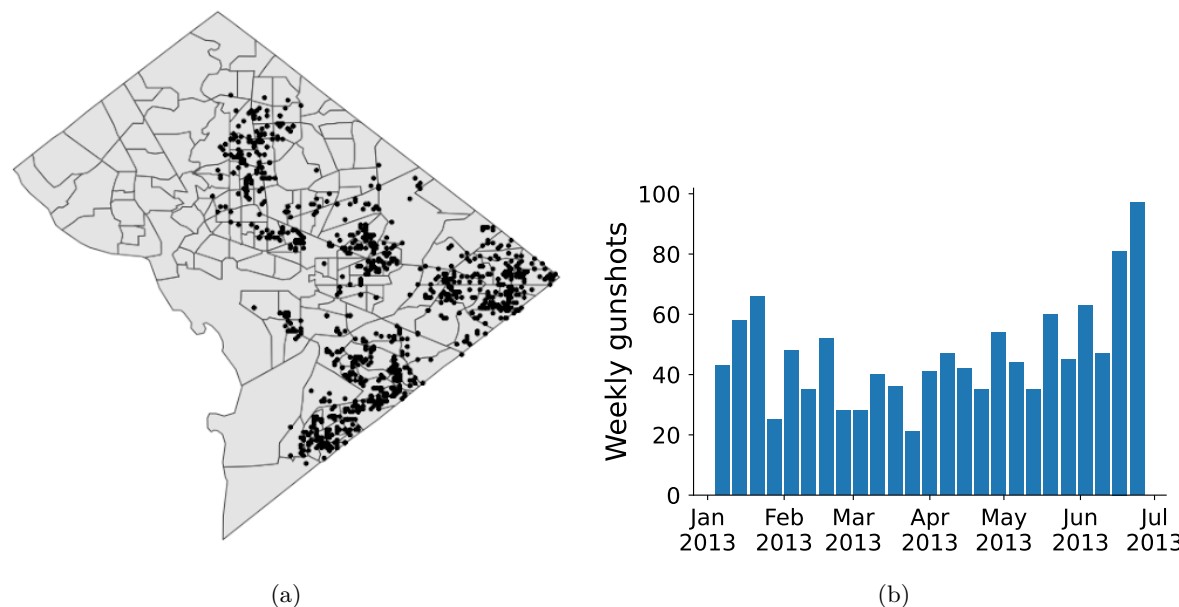

(a)  (b)

Figure 5: Spatial (a) and temporal (b) distribution of the gunfire data in Washington DC over the year 2013. The spatial locations are the exact geographical coordinates and the temporal locations are shown weekly.

We use gunshot data in 2013 recorded by an acoustic gunshot locator system (AGLS) in Washington DC and follow Loeffler & Flaxman (2018) for data preprocessing. There were 1,171 gunshots recorded in total. Spatial locations were rounded to produce approximately 100m spatial resolution and 1 sec temporal resolution. Visualisations of the temporal and spatial distributions of the data are shown in Figure 5(a) and (b).

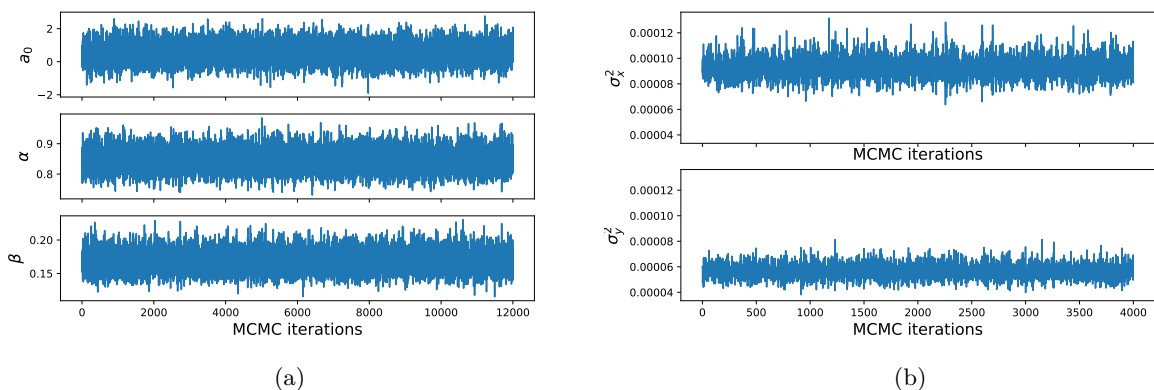

(a)                                                          (b)

Figure 6: MCMC trace for the parameters (a) $a_0, \alpha, \beta$ and (b) $\sigma_x^2, \sigma_y^2$ when collecting the MCMC samples from all chains and discarding warmup.

We perform inference with the HMC (Neal, 2011) routines of numpyro which uses the NUTS (Hoffman & Gelman, 2014) algorithm. We used 2 chains each with $4,000$ samples from which $2000$ are discarded as warmup. We join together the samples from the two chains and report the combined MCMC trace for each of the parameters. Note that we did some prior sensitivity analysis to assess the robustness of our results. We used different parameters on the priors for the parameters and we observed that the posterior distributions of the parameters were similar, giving posterior mean estimates very close to each other. We also tried different priors such as Gamma and Exponential but the convergence of the chains was better when using Truncated Normal distributions. Note that we have rescaled accordingly the temporal and spatial locations of the events, in order to simplify their use in inference and to be appropriate for the domain of our pre-trained GP generators. Rescaling, is standard practice in point process modelling. In Figure 6(a) we report the MCMC trace plots for the parameters $\alpha$ and $\beta$ and $a_0$ and in Figure 6(b) we report $\sigma_x^2, \sigma_y^2$ which define the lengthscales of the spatial Gaussian kernel (x and y distance) that governs excitation in space.

Regarding the Gaussian process fitting, we show the posterior predictive plots in Figures 7 and 8. For the one-dimensional temporal Gaussian process we plot the estimated function $f_t(t)$ in Figure 7. The green line is the mean posterior of $f_t(t)$ and the yellow shaded area is the 90% credible interval obtained from the posterior predictive distribution. The red marks indicate the observed true events.

For the spatial Gaussian process we plot the estimated function $f_{\mathbf{s}}(x, y)$ at the left plot of Figure 8. The mean predictive distribution with the true locations embedded on it is shown on the right. The plots overall

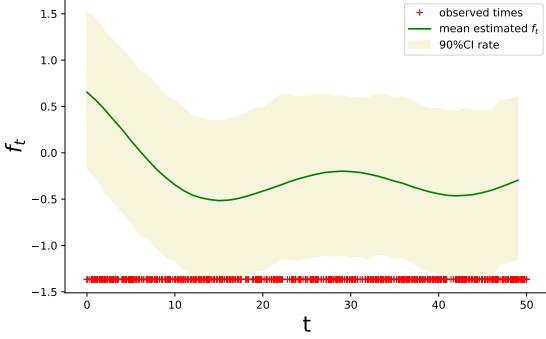

Figure 7: Posterior predictive for $f_t(t)$ with the posterior mean in green and the 90% credible interval in the yellow shaded area, with true time stamps of the events on the x-axis.

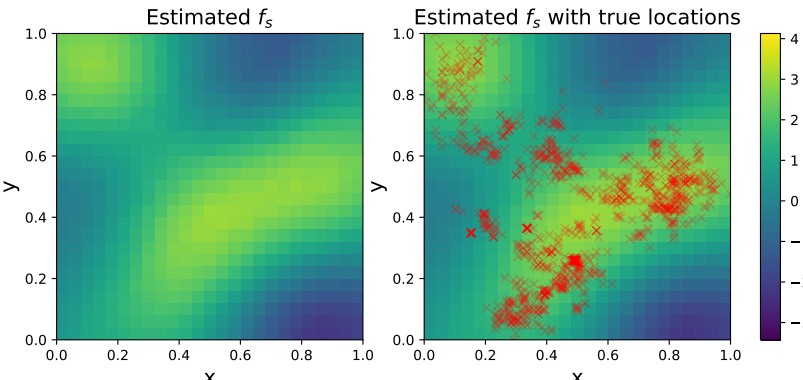

Figure 8: Mean of the posterior predictive for $f_{\mathbf{s}}(x, y)$ (left) and similarly with the true locations (right).

show good convergence with good mixing between the chains and no multimodal behaviour. This is also quantified by the convergence diagnostics $\hat{R}$ which were equal to 1 for all the parameters estimated.

We report an estimate and 90% credible intervals of $\hat{a}_0 = 0.53\,(-0.46, 1.47)$, $\hat{\alpha} = 0.73\,(0.68, 0.78)$, $\hat{\beta} = 0.18\,(0.16, 0.21)$, $1/\hat{\beta} = 5.35(4.64, 6.11)$, $\hat{\sigma}_x^2 = 9.26e^{-5}(7.90e^{-5}, 1.07e^{-4})$, $\hat{\sigma}_y^2 = 5.65e^{-5}\,(4.78e^{-5}, 6.67e^{-5})$ which can be interpreted as below, following the way of interpretation of Loeffler & Flaxman (2018). The average number of shootings triggered by one shooting is around 0.73. Then, rounding to the nearest minute or meter correspondingly, the temporal lengthscale for the exponential triggering kernel is estimated to be around 5 minutes, the spatial triggering lengthscale in $x$ distance, denoted by $\sigma_x$ is around $10m$ and for the $y$ distance $8m$. This means that for every 100 shootings that occur, these create at most another 73. Using the right upper bound of the uncertainty intervals, the period in which diffusion takes place is within less than 6 minutes and the area is within 10 meters in $x$ distance and 8 meters in the $y$ distance. Regarding the background effects the posterior mean of the spatiotemporal Gaussian process is estimated to be 0.53. The results have some differences from the ones reported by Loeffler & Flaxman (2018) and Holbrook et al. (2021) but the model assumed here has a different form and we have applied it on a different subset of the gunshot dataset.

Table 1: Fist row: Average RMSE on test data with its standard error in brackets computed when predicting future unseen temporal and spatial events under the five models. Second row: Average NNL with its standard error in brackets computed when predicting future unseen temporal and spatial events under the five models. Third row: Average NNL with its standard error in bracketss computed on training data under the five models.

|  | Hawkes-LGCP | Hawkes | LGCP | Poisson | DeepSTPP |
|---|---|---|---|---|---|
| RMSE (test) | 7.33 (0.11) | 8.14 (0.13) | 7.90 (0.09) | 14.2 (0.29) | 7.86 (0.25) |
| NNL (test) | -2.534(0.09) | -1.789(0.08) | -0.309 (0.039) | -0.26(0.024) | $-2.24(0.11)$ |
| NNL (train) | $-3.42(0.0007)$ | $-3.15(0.0007)$ | $-2.47(0.0026)$ | $-1.99(0.0003)$ | $-2.76(0.009)$ |

We compare our model to the LGCP model, Hawkes model , baseline Poisson and the state-of-the-art neural network based spatiotemporal model of (Zhou et al., 2022). In our results, we name the model of (Zhou et al., 2022) as DeepSTPP. We report in Table 1 the average RMSE on test data, normalised negative log-likelihood (NNL) on both training and test data. As seen from Table 1 LGCP-Hawkes gives the best predictive performance both in terms of RMSE and NNL. The normalisation of the log-likelihood is done by dividing by the number of training and test data points respectively. The training inference times are

45 minutes for Hawkes-LGCP, 8 minutes for Hawkes, 25 seconds for LGCP, 11 seconds for Poisson and 12 minutes for DeepSTPP.

## 6 Conclusion

We presented a novel model combining Hawkes processes with Gaussian processes, and used it to identify patterns in gun violence in Washington DC. Methodologically, ours is the first model of its kind to have such flexibility in capturing underlying patterns in the rate of occurrence of events, combining a powerful nonparametric statistical model with an interpretable mechanistic self-exciting point process model. This combination means that it can be used across a range of real world spatiotemporal problems in which the underlying data mechanism is unknown. Applications could include social networks, biology, economics and epidemiology. Its general and practical form make it an actionable tool for practitioners that can be used to design interventions and for policy making.

There are many directions for future research. One could study the properties of this model in a theoretical level to define the implications of different forms of the background rate and whether they are identifiable. Additionally, this model can be further extended to include additional covariates. These take the role of marks in a Hawkes process construction and can bring more information in infectious disease applications in which one wants to characterise the disease transmission and quantify the sources that govern infection in space and times. Deviating from a univariate setting, one can consider interacting Hawkes processes to model events in different states or regions where the intensity of events in one regions depends on the intensity in another region. This model could be useful for crime data, and also in neuroscience, where multiple neural trains interact across different parts of the brain. Computationally, this may prove to be a difficult extension. In scenarios where the background trends are potentially coming from different sources, incorporating transformations of Gaussian process could make the framework even more flexible and able to capture multimodal distributions. Finally, one can extend this in a generic flexible framework for Hawkes processes with non-linear intensities that can potentially capture inhibition.

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

# A    Appendix

## A.1    Simulation

In Algorithm 2 we show how to simulate from the LGCP background of the process with intensity $\mu(t, \mathbf{s}) = \exp(f_t(t) + a_0 + f_\mathbf{s}(\mathbf{s}))$. For simulation purposes we write explicitly $a_0 = a_{0t} + a_{0\mathbf{s}}$ but only estimate them via their sum (i.e. $a_0$) as $a_{0t}$ and $a_{0\mathbf{s}}$ are only identifiable through their sum. $p^+$ denotes a probability distribution on the positive real line whereas $p$ denotes a probability on the real line. We take the prior on $\ell_t$ and $\ell_\mathbf{s}$ to be Inverse Gamma, and the one on $\omega_t^2$ and $\omega_\mathbf{s}^2$ to be Log Normal but other options are possible. For the prior on $a_{0t}$ and $a_{0\mathbf{s}}$ we use a Normal distribution.

---

**Algorithm 2** Simulation of the LGCP events from the background

---

**Require:** $T > 0, \mathcal{X}$
    **Draw** $a_{0t}, a_{0s} \sim p(\cdot)$
    **Draw** $l_t, l_\mathbf{s} \sim p^+(\cdot)$
    **Draw** $\omega_t^2, \omega_\mathbf{s}^2 \sim p^+(\cdot)$
    **Form** $k_t, k_\mathbf{s}$ using equation 5
    **Draw** $f_t \sim \mathcal{GP}(0, k_t), f_\mathbf{s} \sim \mathcal{GP}(0, k_\mathbf{s})$
    **Set** $r(t) = \exp(f_t(t) + a_{0t}), r(\mathbf{s}) = \exp(f_\mathbf{s}(\mathbf{s}) + a_{0\mathbf{s}})$
    **Approximate** $I(t, \mathbf{s}) = \int_0^T \int_\mathcal{X} r(t) \times r(\mathbf{s}) \, dt d\mathbf{s}$
    **Draw** $N_0 \sim Pois(I(t, \mathbf{s}))$
    **Draw** $\{t\}_{i=1}^{N_0} \in [0, T]$ from $r(t)$ via rejection sampling
    **Draw** $\{\mathbf{s}\}_{i=1}^{N_0} \in \mathcal{X}$ from $r(\mathbf{s})$ via rejection sampling
**return** $G_0 = (t_i, \mathbf{s}_i)_{i=1}^{N_0}$

---

## A.2    Simulation experiment

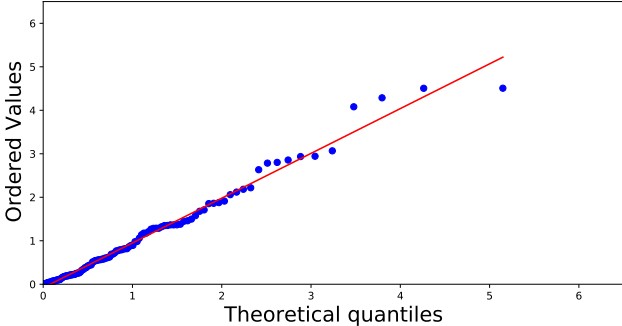

Figure 9: Quantile-quantile plot to compare the quantiles of the empirical and theoretical intensity.

To further support Experiment 1 in section 5.1 we consider goodness of fit tests. We provide a QQ plot and a Kolmogorov-Smirnov test. The QQ plot shown in Figure 9 measures the agreement between the observed and the theoretical quantiles of the intensity function. The theoretical quantiles plotted in x-axis are those from the exponential distribution whereas the empirical ones in y-axis are those from the fitted intensity under our LGCP-Hawkes model. The best fit line on our data has a slope of $1.1$ and intercept $-0.2$. The coefficient of determination is $0.98$ suggesting almost perfect correlation between the two. This suggests a good agreement between the point process model and the experimental data as the points lie on a $45-$degree line. Secondly, we measure the discrepancy between the two using a Kolmogorov-Smirnov test which gives a p value $\gg 0.05$ clearly not rejecting the null hypothesis that the two samples are coming from the same distribution.

