# OpenReview forum: "Cox-Hawkes: doubly stochastic spatiotemporal Poisson processes"
_TMLR — Accepted by TMLR_

### Review · Reviewer_tzF9 · 2022-12-26

**Summary Of Contributions:**

In this submission, the authors proposed an improved spatiotemporal Hawkes process, in which the extrinsic triggering part is modeled by a log-Gaussian Cox process (LGCP). The authors developed an MCMC-based inference method to reduce the computational complexity of the learning process. A cluster generation-based simulation method is developed. This simulation method is a natural extension of the branching process-based simulation method for the classic Hawkes process. Experiments on synthetic and real-world spatiotemporal data show that compared to Hawkes, Poisson, and LGCP, the proposed Hawkes-LGCP model has better performance.

**Audience:**

No

**Claims And Evidence:**

No

**Requested Changes:**

(1) According to the weaknesses above, more state-of-the-art methods should be considered as baselines, and more analytic experiments and more evaluation criteria should be considered.

(2) The advantages of the proposed model and its MCMC-based inference should be demonstrated.


**Strengths And Weaknesses:**

Strengths:
(1) The idea and the description of the proposed method are clear. The derivation of the proposed method is easy to follow.

Weaknesses:
(1) The novelty of the proposed method is limited. The proposed Hawkes-LGCP model is a simple combination of two existing models (i.e., the LGCP model for the exogenous part and the Hawkes model for the endogenous part). The MCMC-based inference is well known as well. The simulation method is a straightforward application of the existing branch processing-based simulation method. In summary, in the aspect of methodology, the contribution of this submission is incremental.

(2) The experimental part is too weak. (a) The baselines include Poisson, classic Hawkes, and LGCP. It is natural that the combination of Hawkes and LGCP (i.e., the proposed method) owns better capacity and performance. However, whether the proposed method can outperform other spatiotemporal point process (STPP) models, e.g., nonlinear Hawkes, self-correcting process, and neural network-based PP models, is not verified. (b) The authors claimed that using MCMC-based inference can reduce the complexity of the learning process. However, the runtime of the proposed method is not tested. Compared to other learning methods, e.g., variational inference, the advantage of the proposed method is not convincing. (c) In the aspect of evaluation, only RMSE is applied. The other commonly-used metrics, e.g., the log-likelihood of testing data, are ignored.

(3) In the aspect of methodology, the proposed model did not consider high-dimensional features of events (which is common in practice). Additionally, for multi-variate STPP cases, especially, those high-dimensional ones, the efficiency of the MCMC-based learning method is questionable. The limitations of the current model and its learning algorithm are obvious and hard to solve.

---

> ### Author Response · Authors · 2023-02-10
> **Response**
>
> We thank the reviewer for their useful and detailed feedback. We give below our response to each part.
>
> (1) Novelty:
> We would like to refer the reviewer to our general comments Point 3 above.
>
> (2) On experiments.
>
> - (a)
> Please see our Points 1 and 2 above.  We also addressed this topic in sections 1 and 2 in the new version.
> - (b) Firstly, we would like note that there is no ‘vanilla’ or classical variational inference (VI)  approach for Hawkes processes. Very recently, (Sulem et al., 2022) introduced a unified framework for variational Bayes inference approaches on multivariate Hawkes processes in purely temporal settings. Similarly, there are neural network approaches for temporal point processes but these do not provide a vanilla VI approach and also do not give interpretable parameters, nor uncertainty quantification as explained above. Our objective is to provide an MCMC approach and not a VI approach, because it guarantee better estimation that is as close as possible to an exact inference while retaining scalability. Recall that VI is targeting an approximate posterior whereas MCMC is targeting the exact posterior. As convincingly demonstrated in (Yao et al., 2018), the variational inference often performs quite poorly at uncertainty quantification. While methods have been proposed to assess the performance of VI (Huggins et al., 2019), performance guarantees do not exist, and workflows to improve VI in specific settings are far from automatic. While our overall model is approximate as it involves an approximate step during inference,  (to sample the GP draws from a pre-trained VAE) our approach is the closest one can get to a full classical exact MCMC approach that retains scalability and practicality. On the other hand, a classical MCMC approach with our objectives (space+time+clustering+contagion) is very costly and not practical. Finally using the capabilities of the probabilistic language numpyro, we are able to provide code for the MCMC approach. For the purposes we have outlined above, we prefer to focus on an MCMC approach rather than VI like other similar work in literature (Lindreman and Adams, 2014, Loeffler and Flaxman, 2018,  Mohler, 2013).
>
> Regarding runtime: We added them in section 5.3.
>
> D. Sulem, V. Rivoirard, and J. Rousseau, Scalable Variational Bayes methods for Hawkes processes, arXiv:2212.00293, 2022.
> Y. Yao, A. Vehtari, D. Simpson, and A. Gelman,  Yes, but Did It Work?: Evaluating Variational Inference, ICML, 2018.
>
> J. Huggins , M. Kasprzak ,T. Campbell, and T. Broderick, Validated Variational Inference via Practical Posterior Error Bounds, AISTATS 2020.
>
>  S. Linderman, C. Stock., and R. Adams. A framework for studying synaptic plasticity with neural spike train data. NeurIPS, 2014.
>
> C. E. Loeffler  and S. R. Flaxman, Is gun violence contagious? a spatiotemporal test. Journal of Quantitative Criminology, 34:999–1017, 2018.
>
> O. G. Mohler,  Modeling and estimation of multi-source clustering in crime and security data. The Annals of Applied Statistics, 7(3):1525–1539, 2013.
>
> Z. Zhou, X. Yang, R. Rossi, H. Zhao and R. Yu, Neural Point Process for Learning Spatiotemporal Event Dynamics. In Proceedings of The 4th Annual Learning for Dynamics and Control Conference, 2022.
>
> - (c)
>
> The negative normalised log-likelihood of the experiments is now added in the paper for synthetic data in section 5.2 and on the real gunshot dataset in section 5.3.
>
> (3) On methodology
>
> Regarding the MCMC-based learning method, it may be true that its efficiency is questionable for simple random walk proposals, but not for the current state-of-the-art gradient based samplers such as Hamiltonian Monte Carlo. See for example Izmailov et al., 2021 where extremely high dimensional neural networks are fitted using HMC. Furthermore, as we have noted above, variational inference, which may be faster, is not a gold standard for Bayesian inference. To reiterate, our goal in this paper was to introduce a new model to represent natural phenomena and to reliably perform inference over this model. We believe we have demonstrated this. We hope for faster, more efficient algorithms in the future but that is out of the scope of this work.
>
> We agree that an MCMC may be challenging in a case with multidimensional features or multivariate processes but this is not the scope of our paper nor the objective of our proposal. We are proposing a linear univariate Hawkes process model with a flexible intensity that can capture arbitrary trends in the background for spatiotemporal clustering and also contagion effects from self-excitation. A lot of practical settings such as crime data, social interactions, epidemiological spreading are not high dimensional. Our model can cover these cases while considering both temporal and spatial locations.
>
> P. Izmailov, and S. Vikram, and M. D. Hoffman, and A. Gordon, and G. Wilson, What Are Bayesian Neural Network Posteriors Really Like? ICML, 2021.

---

### Review · Reviewer_3Wae · 2023-01-17

**Summary Of Contributions:**

The authors propose a variant of a spatio-temporal Hawkes process which combines a Hawkes process for inducing mutually exciting effects and a Gaussian process for capturing clustering effects in the background rate . Based on that they provide some synthetic experiments for demonstrating their model. They also test their model on the gunshot data. Their model outperforms simpler baseline point process models in terms of RMSE in the event predictions.

**Audience:**

Yes

**Broader Impact Concerns:**

no concerns

**Claims And Evidence:**

Yes

**Requested Changes:**

**(1) Some references should be included.**

It would strengthen the paper if the authors also compare with similar (mathematically or functionally ) similar models [1] [2]. Especially [2] is a very similar model. The authors extend [2] to include the space domain while adding a Hawkes process intensity. However, the use of a GP in the intensity of a point process detracts from the mathematical novelty of the article.

Moreover, one of the main motivation of authors work is the lack of expressive enough background rates. However, the intensity of [2] offers such an expressive background rate. Similarly, in [3] the background rate is a constant term multiplied by a probability term capable of modeling complex (inhibitory) effects. Finally, when the authors refer to the cluster-based Hawkes process simulation they should cite [5]


**(2) Some points in the presentation should be corrected.**

a. Section 3.1 should be renamed to regular point process definition. The definitions of this section refer to the broad class of point processes characterized by an intensity function and not to a Hawkes process. Moreover, it should be stated explicitly that $\lambda \ge 0$. There is also a typo: $B(s,s)$ should be $B(s, \Delta s)$ in the explanatory text below equation 1.

b. In Algorithm 1:
the condition $G_l!=0$ does not appear correct since $G_l$ in the last line of the body of the loop seems to increase (to include all the generated events so far. I understand that the authors want to convey: unless there are no more generated events in the time window [0,T] . However, they should accurately formulate this. Moreover, in the simulation steps ( $C_i$, $O_i$), they should provide the exact distributions they are sampling from. Finally the notation ${t,s} \sim \mu(t,s)$ is also not correct since $\mu$ refers to the background rate and not a distribution. Probably the distribution is a Poisson process characterized by $\mu(s,t)$?


**(3) Some technical details are missing:**

The authors should provide the exact MCMC updates similar to [6]. Regarding the identifiability results, the authors mention:

In purely temporal cases we have guarantees on the estimationand under certain conditions we have consistency results as well identifiability results. However once we addthe spatial component, the results do not necessarily extend. Therefore, we take here a separable form of aproduct of an exponential kernel in time and a Gaussian kernel in space.

I think the authors should either cite accordingly or provide explicit derivations for supporting this statement.

**(4)In the synthetic experiments, the model should be further validated**

a. It would be helpful if the authors should also provide goodness-of-fit tests (Kolmogorov-Smirnov), see [7], [3]

b. clarification: in the synthetic examples, is the VAE-based GP approximation used? In either case, showing similar results for both cases could help justify the use of a VAE-based GP approximation in the point process context.

c. In figure 4, it would help if authors also provide the mean values of the inferred parameters. Ideally, a LGCP-Hawkes should correctly recover either a pure Hawkes (zero covariance in the GP?) or a LGCP (zero excitation coefficients). However, there is a discrepancy in the error in these cases.

d. a visualization of the GP background rate could provide further insights.

**(5)In the gunshot experiments, some details are missing**

a. Can the authors provide inference time?

b. I also think the authors should provide the loglikelihood (not only specific parameters) to demonstrate convergence of their inference algorithm

c. The authors use truncated normal priors. Gamma or exponential priors are also commonly used. Demonstrating sensitivity wrt to the prior type would also strengthen the paper.






[1] Chen, Ricky TQ, Brandon Amos, and Maximilian Nickel. "Neural Spatio-temporal point processes.

[2] Malem-Shinitski N, Ojeda C, Opper M. Flexible Temporal Point Processes Modeling with Nonlinear Hawkes Processes with Gaussian Processes Excitations and Inhibitions.

[3] Apostolopoulou, I., Linderman, S., Miller, K., & Dubrawski, A. (2019). Mutually regressive point processes. Advances in Neural Information Processing Systems, 32.

[4] RUBIN, Izhak. Regular point processes and their detection. IEEE Transactions on Information Theory, 1972, 18.5: 547-557.

[5]. G. Hawkes and D. Oakes, “A cluster process representation of a self-exciting process,”
Journal of Applied Probability, vol. 11, no. 3, pp. 493–503, 1974.

[6] RASMUSSEN, Jakob Gulddahl. Bayesian inference for Hawkes processes. Methodology and Computing in Applied Probability, 2013, 15.3: 623-642.

[7] Emery N Brown, Riccardo Barbieri, Valerie Ventura, Robert E Kass, and Loren M Frank. The time- rescaling theorem and its application to neural spike train data analysis. Neural computation, 14(2):325–346, 2002.



**Strengths And Weaknesses:**

**Strengths**

In general, it is an easy-to-read paper. The visualizations in the paper also help the reader comprehend the functionality of the model. I also appreciate the source code the authors are providing. The provided experiments show promise for use of their model inI w real-world applications.

**Weaknesses**

I will expound more below. In general:

(1) some points in the presentation should be corrected.

(2) both the introduction and the experiments should be expanded to include relevant work.

(3) Some inference details ( metropolis hasting ratios), simulation details, and identifiability results should be explicitly stated to make the article self-contained (without having to resort to the source code for mathematical details).

---

> ### Author Response · Authors · 2023-02-10
> **Response**
>
> We thank the reviewer for their useful and detailed feedback. We give below our response to each part.
>
> (1) Some references should be included.
>
> We have now included references [1] (Chen et al.,  2021), [2] (Malem-Shinitski et al., 2021) and [3] (Apostolopoulou et al., 2019) in the new version of the paper in section 1. Also, we cite [5] (Hawkes and Oakes,1974)  in section 4.1. Regarding a potential experimental comparison of our model to [1,2,3]  we refer the reviewer to our response above on Point 1: Comparison with Neural Network Models and Point 2: Comparison with other Point Process Models.
>
> (2)  Some points in the presentation should be corrected.
> We have made the suggested changes in section 3.1 the new version.
>
> (3) Some technical details are missing
> - a. To implement our approach we use a Hamiltonian Monte Carlo (Neal, 2011) using the NUTS algorithm (Hoffman and Gelman, 2014) implementation in numpyro. We propose an MCMC approach with a novel way to sample GP draws, but we do not give a new type of MCMC. Our generative models are given within the framework of numpyro. Then, the updates are the NUTS sampler updates.
>
>  R. N. Neal, MCMC Using Hamiltonian Dynamics, Handbook of Markov Chain Monte Carlo, 113-162, 2011.
>
>  M. D. Hoffman and  A. Gelman, The No-U-turn sampler: adaptively setting path lengths in Hamiltonian Monte Carlo, Journal of Machine Learning Research, 15:1593-1623, 2014.
>
> - b. Regarding the comment on theoretical guarantees, we have now added an extra paragraph in the new version in section 3.2 to properly explain and cite the results we are referring to. For completeness we also add here this information.
> In purely temporal cases there exist guarantees on the estimation and under certain conditions there are consistency results as well identifiability results. For multivariate linear purely temporal Hawkes processes with constant background it is a known result that one can recover the parameters, i.e. the process is identifiable and one can also prove consistency results. Donnet et al., 2020 prove this and give posterior concentration rates. Some results exist for non-linear Hawkes processes with constant backgrounds: Bremaud and Massoulie, 1996 provide results on the uniqueness of the stationary solution but they do not study estimation of the parameters. Similarly, Sulem et al., 2021 study general non-linear and nonparametric Hawkes processes and provide conditions on the Bayesian methods to estimate the parameters with a (presumably optimal) concentration rate. Other approaches with time-varying backgrounds exist (e.g. Unwin et al., 2021, Zhou et al., 2020) but there are no theoretical results that apply directly in that case (linear or non-linear). Note though that for any Hawkes process with a temporally varying background, stationarity is not relevant anymore as the background is changing in time and therefore the expectation of the intensity cannot be constant.
>
> S. Donnet, V. Rivoirard, and J. Rousseau, Nonparametric bayesian estimation of multivariate hawkes processes. The Annals of Statistics, 48(5):2698–2727, 2020.
>
> P. Brémaud and L. Massoulié, Stability of nonlinear hawkes processes. Ann. Probab., 24(3): 1563–1588, 1996.
>
> D. Sulem, V. Rivoirard, and J. Rousseau, Bayesian estimation of nonlinear hawkes process. arXiv.2103.17164, (5), 2021.
>
> H. J. T. Unwin, I. Routledge, S. Flaxman, M-A. Rizoiu, S. Lai, J. Cohen, D. J. Weiss, S. Mishra, and S. Bhatt, Using hawkes processes to model imported and local malaria cases in near-elimination settings. PLOS, 2021.
>
> F. Zhou, Z. Li, X. Fan, Y. Wang, A. Sowmya, and F. Chen, Efficient inference for nonparametric hawkes processes using auxiliary latent variables. JMLR, 21:1–31, 2020.
>
>
> (4) In the synthetic experiments, the model should be further validated
> - a.  We add goodness of fit tests in section 5.1 with details in Appendix A2
> - b. Yes the VAE-based GP approx. is used in experiments. A non-VAE approach is very costly as the inclusion of an LGCP background is not trivial.
> - c. We added and interpreted the parameter estimates and other useful quantities obtained from the intensity in section 5.2 in the new version.
> - d. In Figures 1 and 2 we plot the background GP separately for time and space (as in Figures 7,8). We are unaware of useful ways to plot the background rate for both time and space.
>
> (5)
> - a. We have provided them in section 5.3 of the new version.
> - b We now provide in section 5.3 the mean estimate and standard error of the normalised negative log-likelihood on both training and test data for all the competing models. We note that we are performing Bayesian inference and convergence of just the log likelihood is not as relevant or informative as the trace of the parameters.
> - c We have also tried Gamma and exponential priors but the mixing of chains was not good, whereas truncated normals worked better. We have also mentioned this in section 4.2 of the new version of the paper to address this concern.

---

### Review · Reviewer_qAT6 · 2023-01-28

**Summary Of Contributions:**

The authors propose the Hawkes-LGCP point process model for timestamped events taking place at different locations in space. It uses a Hawkes process with triggering mechanisms for both time and space to allow past events to encourage future events nearby in space and in the near future. It further incorporates a log-Gaussian Cox process (LGCP) prior for the Hawkes process background rate. The authors apply their model to analyze a data set of gunshots in Washington DC and find that one gunshot typically triggers 0.7 future shots and within a roughly 10x8 meter area in the next 6 minutes.

The main contributions I observe are as follows:
- Novel formulation for a self-exciting spatiotemporal point process by incorporating an LCGP prior for the background rate in a spatiotemporal Hawkes process.
- Efficient algorithms to sample from this Hawkes-LGCP model and to perform Bayesian inference.
- Interesting analysis on gunshot data that attempts to identify how much influence a gunshot has on future gunshots nearby.

**Audience:**

Yes

**Broader Impact Concerns:**

No broader impact statement is provided, and I think one is necessary. See my discussion in the Weaknesses section.


**Claims And Evidence:**

Yes

**Requested Changes:**

Major issues:
- Section 3.2, 3rd last paragraph: "$N(t) - N(\mathbf{s}) = \infty$ for $t - \mathbf{s} < \infty$.": This is confusing because $\mathbf{s}$ has been used to represent space and is a vector. In this equation, I believe $s$ should be a scalar representing time, so I suggest a different variable name. Furthermore, I am not sure if this paragraph fits given the previous paragraph. The previous paragraph is expressing almost the same content but more specific to the exponential kernel and Gaussian form taken in this paper.
- Section 4.1, last paragraph: Provide more details (in an appendix) or at least a reference on how to sample from the LGCP by drawing an approximate realization then using rejection sampling.
- RMSE computation for Figure 4: clarify whether the RMSE is just over the (x,y) spatial locations or whether it includes error in timestamp as well.
- Either specify the rescaling that was performed on time and space in the gunshot data set so that the reader can better interpret the parameter estimates (e.g., how large of a range does $\sigma_x^2 = 9.26e^{-5}$ really correspond to?). Alternatively reverse scale the parameter estimates back into human interpretable units, such as meters for distance and minutes for time.

Minor issues:
- All references that are published in conference proceedings seem to be missing the name of the proceedings, e.g. Linderman & Adams (2014), which was published in the ICML proceedings.
- Equation references seem incorrectly formatted, e.g. "Eq equation 7" and "Eq equation 8".
- Page 2, second last paragraph: "prior on t he background": t he -> the
- Section 3.1, first paragraph: "the appearance of future events. s": Remove extra "s".
- Section 3.1, first paragraph: "$|B(\mathbf s, \mathbf s)|$ is the Lebesgue measure of the ball $B(\mathbf s, \mathbf s)$ with radius $\mathbf s > 0$.": I suggest using different letters for the center of the ball and the radius, e.g. $\mathbf s$ and $\Delta \mathbf s$, as in equation (1).
- Section 3.2, paragraph after equation (2): However, we The condition -> However, the condition
- In Section 3.4, the spatial coordinates now switch to $s$, not $\mathbf s$. Stay consistent in your notation.
- Section 4.2, second paragraph after equation (9): linearas -> linear as
- Section 4.2, second paragraph after equation (9): lgcp -> LGCP
- Figure 2 is blurry and should be created in vector format or higher resolution raster.



**Strengths And Weaknesses:**

Strengths:
- LCGP prior for the background allows for a fully generative process that can be used to simulate from Hawkes processes with a variety of background rates compared to kernel density estimation approaches (e.g. Loeffler & Flaxman, 2018).
- Model is very flexible and improves prediction accuracy, yet the parameters are still very interpretable, making it useful for exploratory analysis.

Weaknesses:
- Many (mostly minor) presentation issues scattered throughout the manuscript. It almost feels like it has never been proofread. I have tried to point out most of these below.
- Broader impacts of the proposed work should be discussed. This is particularly important because the authors analyze a gunshot dataset to draw potential conclusions about gunshots triggering other gunshots. Their conclusions may have implications for public policy, so limitations of their conclusions need to be made very clear.

---

> ### Author Response · Authors · 2023-02-10
> **Response**
>
> We thank the reviewer for their useful and detailed feedback. We give below our response to each part.
>
> Major Issues:
>
> - We agree that the use of s is confusing and we changed this in section 3.2 the new version. We now make sure that we use s only to denote space throughout the paper. We have also revised the paragraph the reviewer is referring to.
> - We have added an algorithm (Algorithm 2) in appendix A.1 in the new version.
> - The RMSE considers both timestamps and spatial locations. We have provided more explanations and a definition of our metric in the new version in section 5.2.
> - As mentioned in the paper in section 5.3, we follow Loeffler and Flaxman, 2018 to preprocess the data. Specifically, we rescale the data as exactly was performed in Loeffler and Flaxman, 2018 whose code can be found at the github repository https://github.com/flaxter/gunshot-contagion in script data.r
> In section 5.3. towards the end we interpret the parameters in terms of seconds (or minutes) and meters. “...rounding our time parameters to the nearest minute and to the nearest meter our spatial parameters, the temporal lengthscale for the exponential triggering kernel is estimated to be around 5 minutes, the spatial triggering lengthscale for the x distance, denoted by sigma_x is around 10m and for the y distance is 8m. This means that for every 100 shootings that occur, these create at most another 73. Using the right upper bound of the uncertainty intervals, the period in which diffusion takes place is within less than 6 minutes and the area is within 10 meters in x distance and 8 meters in y distance.”
>
>
> Minor Issues:
> - We thank the reviewer for pointing these out and we have revised them all in the new version.

---

> > ### Comment · Reviewer_qAT6 · 2023-03-12
> > **One more minor issue**
> >
> > I have read the authors' responses and looked through their revision. I believe that most of the important issues raised by the reviewers have been addressed.
> >
> > I do agree with another reviewer's comment regarding limited novelty and significance. However, TMLR guidelines specifically state not to use these as reasons to recommend rejection. I do believe that some individuals in TMLR's audience would be interested in the findings of this paper, so that is sufficient novelty and significance, in my opinion.
> >
> > I noticed one new minor issue introduced in the revision: Table 1 caption: "Fist row" -> "First row"

---

### Author Response · Authors · 2023-02-10
**Some General comments (1/2)**

We would like to thank the reviewers for their time and their feedback on our manuscript. We have taken their suggestions into account and have revised our paper. Below, we reply to each reviewer individually to address their comments. Here we address two common points raised by reviewers tzF9 and 3Wae.

Point 1: Comparison with Neural Network Models

Most of the proposed (and suggested in the review) neural network models (Du et al., 2016; Mei and Eisner, 2016; Omi et al., 2019; Zhang et al., 2022) are only temporal in nature and there exists no straightforward or elegant way to extend them to spatiotemporal settings. Few of them provide a way to model space by treating spatial locations as discrete ‘marks’. However, the use of marks doesn’t allow us to model spatial correlations, as observed in many real-world settings. Okawa et al., 2019 extend neural networks to spatial settings but lack the ability to predict the next event in space and time. More recently, Chen et al., 2020 and Zhou et al., 2022 (23--24 Jun 2022) provide a way to model spatiotemporal processes with neural networks. However, both of them still lack the ability to capture a more flexible background setting where background intensity is not just a constant but by itself produces clustering in time and space (we like to emphasise here that this clustering is different from the one obtained from the self-exciting or inhibiting nature of the process). Moreover, Chen et al., 2020, use a neural ODE framework which is very computationally expensive, making it impractical in many application settings. Hence in our new set of experiments (new table provided in Table 1 in section 5.3 in the new version), we provide a comparison against Zhou et al., 2022 which we call DeepSTPP and is currently the state-of-the-art for the neural network spatial-temporal point process models.

|             | Hawkes-LGCP    | Hawkes         | LGCP           | Poisson       | DeepSTPP      |
|-------------|----------------|----------------|----------------|---------------|---------------|
| RMSE (test) | 7.33 (0.11)    | 8.14 (0.13)    | 7.90 (0.09)    | 14.2 (0.29)   | 7.86 (0.25)   |
| NNL (test)  | -2.534(0.09)   | -1.789(0.08)   | -0.309 (0.039) | -0.26(0.024)  | -2.24 (0.11)  |
| NNL (train) | -3.42 (0.0007) | -3.15 (0.0007) | -2.47(0.0026)  | -1.99(0.0003) | -2.76 (0.009) |



Point 2: Comparison with other Point Process Models

We agree with the reviewers that there exist other types of point processes such as non-linear Hawkes (Malem-Shinitski et al., 2022; Sulem et al., 2021) and mutually regressive point processes (Apostolopoulou et al., 2019) to name a few. As stated most of them are just temporal, and extending them to spatial settings is non-trivial and hence these are unsuitable for a direct comparison with our model. For completeness, we have now included the above references in section 1 of the updated version of our paper. Finally, we note here that our proposal accounts for the combination of a flexible background with a self-excitation (contagion) part in the Hawkes process intensity. Therefore, any subsequent new approaches that improve upon the contagion part can be easily incorporated into our model.



H. N. Du, R. Dai, U. Trivedi, M.  Upadhyay, M. Gomez-Rodriguez, and L. Song, Recurrent Marked Temporal Point Processes: Embedding Event History to Vector. ACM SIGKDD, 2016.

H. Mei. and J. Eisner, The neural Hawkes process: A neurally self-modulating multivariate point process. NeurIPS 2017.

L-N. Zhang, J-W. Liu,  Z-Y. Song, and X. Zuo, Temporal attention augmented transformer Hawkes process. Neural Comput. Appl. 34, 3795–3809, 2022.

T. Omi, N. Ueda, and  K. Aihara, Fully neural network based model for general temporal point processes. NeurIPS, 2019.

M. Okawa, T. Iwata, T. Kurashima, Y. Tanaka, H. Toda, and N. Ueda. Deep Mixture Point Processes: Spatio-temporal Event Prediction with Rich Contextual Information.  ACM SIGKDD, 2019.

R. T. Q. Chen, B. Amos, and M. Nickel, Neural Spatio-Temporal Point Processes. ICLR, 2021.

Z. Zhou, X. Yang, R. Rossi, H. Zhao and R. Yu, Neural Point Process for Learning Spatiotemporal Event Dynamics. L4DC, 2022.

N. Malem-Shinitski, C. Ojeda and M. Opper, Flexible temporal point processes modeling with nonlinear Hawkes processes with Gaussian processes excitations and inhibitions. ICML, 2021.

D. Sulem, V. Rivoirard, and J. Rousseau. Bayesian estimation of nonlinear Hawkes process. arXiv:2103.17164, 2021.

N. Malem-Shinitski, C. Ojeda, and M. Opper, Variational Bayesian Inference for Nonlinear Hawkes Process with Gaussian Process Self-Effects. Entropy  24, 356, 2022.

I. Apostolopoulou, S. Linderman, K. Miller, and A. Dubrawski, Mutually regressive point processes. NeurIPS 2019.

---

### Author Response · Authors · 2023-02-10
**Some general comments (2/2)**

We would like to thank the reviewers for their time and their feedback on our manuscript. We have taken their suggestions into account and have revised our paper. Below, we reply to each reviewer individually to address their comments.
Here we we would like to reiterate why this work is novel and significant. This was a concern of reviewer tzF9.

Point 3: On novelty and significance of the work.

Novelty: To our knowledge there is no other paper in the literature that has attempted to combine together log-Gaussian Cox Processes and Hawkes processes in continuous time; we do this in a Bayesian and spatiotemporal setting. Combining these two modelling mechanisms in an explainable way that covers both space and time is not only novel but also significant and highly non-trivial.

Significance: Our approach allows us to explain the appearance of events along with dependencies between them by not limiting ourselves to the cases where there is pure contagion or pure clustering. For count data, in spatial statistics applications, researchers use Cox processes whereas, for temporal data, researchers use Hawkes processes. It is highly likely that real word data requires both these mechanisms and likely more to explain the data generating mechanism. We allow events to be explained by both spatial and temporal clustering due to background effects along with excitation(contagion) through historic events. These two mechanisms combined in a single model with directly interpretable parameters gives us a unique way to explain clustering and contagion patterns in spatiotemporal data to be then used in social or epidemiological settings.

Non-trivial: bringing clustering and triggering mechanisms together is non-trivial. The simulation and inference are costly due to the cubic complexity in Gaussian Process operations (inversion, determinant etc), and the quadratic complexity from computing the likelihood of the self-exciting kernel function. Our approach is new, making use of novel recent VAE capabilities to scale up statistical inference. We circumvent the difficulties arising from a vanilla MCMC approach by using pre-trained VAEs (Semenova et al., 2022) that give us cheap access to Gaussian Process samples during the MCMC sampling routine.

E. Semenova, Y. Xu, A. Howes, R. Theo, S. Bhatt, S Mishra, and S. Flaxman. Priorvae: encoding spatial priors with variational autoencoders for small-area estimation. Royal Society Publishing, 73–80, 2022.

---

### Decision · Action_Editors · 2023-04-11

**Recommendation:** Accept as is

**Comment:**

I am happy to recommend acceptance of this paper, concurring with the opinion of the three reviewers. The paper presents a new approach for point process modeling based on spatiotemporal Hawkes processes with an underlying log-Gaussian cox process. While this does combine well-studied components, the combination proves clearly useful and the proposed inference algorithm advances the literature in this area. The changes made in response to the reviewers has addressed their remaining concerns regarding run-time and comparisons, and the re-write has improved the overall presentation.

The reviewers still believe it would be worthwhile comparing against Chen et al, since some of the datasets they use seem comparable in size with the gunshot data used in this paper. However, while I encourage the authors to include this, I do not think it should be a prerequisite for acceptance, since the paper does include comparison with the more recent Zhou et al paper (which itself shows improvement over the Chen et al paper).


**Audience:**

Yes, the paper is of interest to members of the TMLR audience working on Gaussian processes, point processes, and Bayesian methodology more generally.

**Claims And Evidence:**

The claims are well-supported empirically using both synthetic and real data. Neither I nor the reviewers have any concerns about the validity of the sampler or the model.